# The role of the physical environment in stroke recovery: Evidence-based design principles from a mixed-methods multiple case study

Ruby Lipson-Smith[1]*, Heidi Zeeman[2], Leanne Muns[3], Faraz Jeddi[4], Janine Simondson[5], Julie Bernhardt[1]

1 The Florey Institute of Neuroscience and Mental Health, Heidelberg, Victoria, Australia, 2 Menzies Health Institute Queensland, Griffith University, Gold Coast, Queensland, Australia, 3 Bendigo Health, Clinical Operations, Bendigo, Victoria, Australia, 4 Bendigo Health, Department of Integrated Medicine, Bendigo, Victoria, Australia, 5 St Vincent's Hospital Melbourne, Physiotherapy and Rehabilitation Unit, St George's Hospital, Kew, Victoria, Australia

* ruby.lipson-smith@florey.edu.au

**Data Availability Statement:** All quantitative data files are available from the melbourne.figshare.com database (https://doi.org/10.26188/21399891). All

## Abstract

Hospital design can impact patient outcomes, but there is very little healthcare design evidence specific to stroke rehabilitation facilities. Our aim was to explore, from the patient perspective, the role of the physical environment in factors crucial to stroke recovery, namely, stroke survivor activity (physical, cognitive, social), sleep, emotional well-being, and safety. We conducted a mixed-methods multiple-case study at two inpatient rehabilitation facilities in Victoria, Australia, ($n = 20$ at Case 1, $n = 16$ at Case 2) using "walk-through" semi-structured interviews, behavioural mapping, questionnaires, and retrospective audit. Four interrelated themes emerged: 1) entrapment and escape; 2) power, dependency, and identity in an institutional environment; 3) the rehabilitation facility is a shared space; and 4) the environment should be legible and patient-centred. Quantitative data revealed patterns in patient activity; stroke survivors spent over 75% of their time in bedrooms and were often inactive. Convergent mixed methods analysis was used to generate a new conceptual model of the role of the physical environment in stroke survivors' behaviour and well-being, highlighting the importance of variety and interest, privacy without isolation, and patient-centred design. This model can be used by designers, healthcare providers, and policy makers to inform the design of rehabilitation environments.

## Introduction

The physical environment of hospital buildings can influence clinical outcomes, patient experiences, safety, efficiency, and cost [1, 2], but there is no 'one-size-fits-all' for healthcare design; buildings must meet the needs of the patients, staff, and clinical procedures that they are designed to serve. Healthcare services are complex, multifactorial systems with many interconnected components–including the physical environment, social environment, and the individual occupants–that interact and adapt in often unpredictable ways [3–6]. This paper presents an exploration of one such complex system–stroke rehabilitation environments–with the aim

relevant qualitative data are within the paper and its Supporting Information files.

**Funding:** "RLS was supported by a Research Training Program PhD scholarship from the Australian federal government. JB is supported by a NHMRC Fellowship (RF1058635). The Florey Institute of Neuroscience and Mental Health acknowledges the support from the Victorian government and in particular the funding from the Operational Infrastructure Support Grant. The funders had no role in study design, data collection and analysis, decision to publish, or preparation of the manuscript."

**Competing interests:** The authors have declared that no competing interests exist.

to develop a conceptual model to inform the physical design of purpose-built facilities for inpatient stroke rehabilitation.

After receiving acute care, stroke survivors are often transferred to a sub-acute inpatient rehabilitation facility to receive personalised rehabilitation therapy. In contrast to patients in acute care, rehabilitation patients are not passive receivers of care; rather, they must be engaged and active participants, while balancing activity with risk to their safety (e.g., falls). Clinical guidelines for stroke rehabilitation recommend that stroke survivors participate in targeted, goal-directed practice in and out of therapy, as well as additional general physical and cognitive activity to promote their learning and recovery post-stroke [7, 8]. Rest and sleep post-stroke are also important [9, 10], as is emotional well-being–it has long been recognised that stress, depression, and low mood are deleterious for stroke recovery [11], and patients feel that their practice and activity is hampered by boredom, lack of motivation, and perceived lack of autonomy [12].

These specific needs of inpatient stroke rehabilitation have previously been largely overlooked in healthcare environments research [13, 14]. Rehabilitation facilities are rarely purpose-built and, despite supporting very different activities, they often physically resemble acute care facilities [15]. It is therefore perhaps unsurprising that observed behaviour and mood of individuals in inpatient rehabilitation is often not optimal for recovery. Stroke survivors in inpatient rehabilitation are largely inactive and alone outside of their scheduled therapy time [16–18], and time in therapy is likely to be less than what is recommended [8] or required to support recovery [19]. Post-stroke depression and anxiety are common [20, 21], many stroke patients report feeling bored while in hospital [22], and lack the motivation and autonomy to participate in practice [12]. We argue that purpose-built environments that support the specific needs of stroke rehabilitation are needed.

We previously developed an expert-informed framework of what is important in the physical environment for optimal stroke rehabilitation [23], where the 'physical environment' includes architectural and landscape features, interior design features, ambient features (e.g., noise, light, air quality), and maintenance and cleanliness [24]. In the present study, we build on our previous work by exploring, from the patients' perspective, the role of the physical environment in identified priority objectives, namely, patient behaviour (activity and rest), emotional well-being, and safety.

In healthcare environments research, the interdependencies between the physical environment, the social environment, and the patient present a challenge for traditional scientific approaches of isolating variables and identifying causation [25]. Complex systems like healthcare settings are best understood when the system is observed as an integrated whole, without attempts to control or simplify any aspect of it [26, 27]. In the present study, we addressed the following research questions: What are stroke survivors' experiences of the physical environment of inpatient rehabilitation facilities? What is the role of the physical environment of these facilities in stroke survivor behaviour (activity and rest), emotional well-being, and safety? We took the position that connections and interdependencies between the patient and the environment in stroke rehabilitation can only be identified if they are observed in context. We therefore chose to use a case study design to explore the role of the physical environment in stroke survivor behaviour, emotional well-being, and safety. We planned to express our case study findings in a clear, conceptual model which architects and planners could use in their practice to inform the design of purpose-built inpatient stroke rehabilitation facilities.

## Materials and methods

### Study design and setting

This study was a patient-focused, convergent mixed-methods multiple-case study [28–30], called the 'ENVironments for Inpatient RehabilitatiON of Stroke patients' (ENVIRONS)

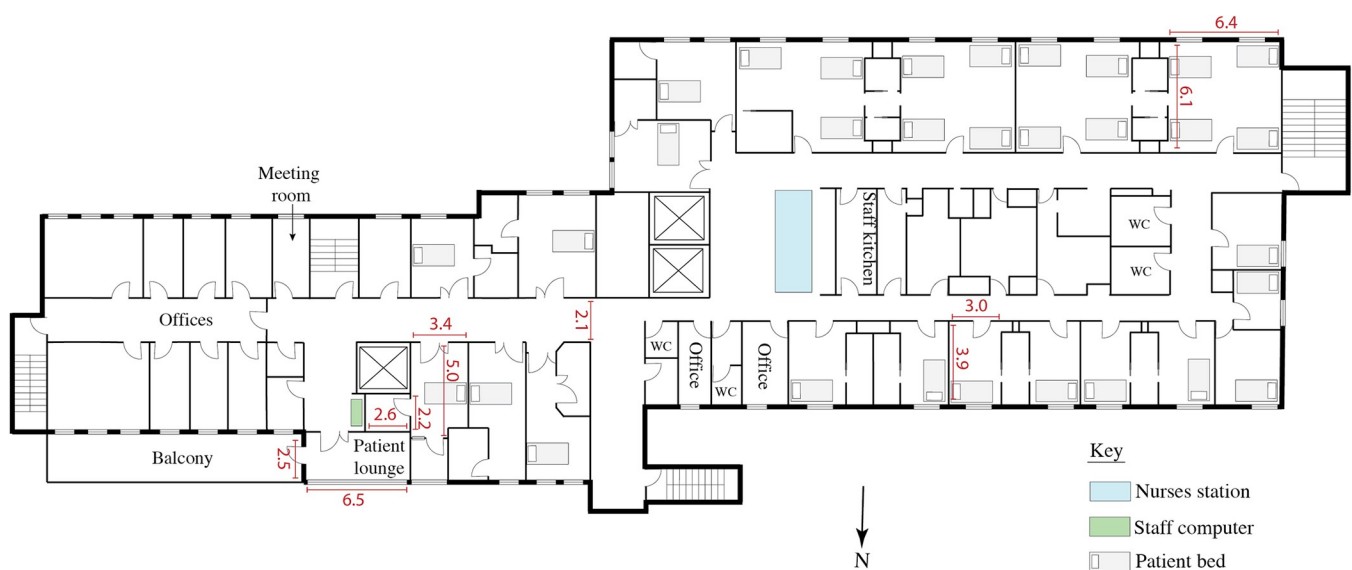

**Fig 1. Floor plan of the inpatient rehabilitation ward at Case 1.** The ward is located on the 1st floor. The gym and other therapy areas are on the ground floor. All measurements are in meters. Plan is not to scale.

study. Our data collection and analysis focused on the role of the physical environment in stroke survivor activity, rest, emotional well-being, and safety. The cases were two inpatient rehabilitation facilities in Victoria, Australia: [removed for peer review] (Case 1) and [removed for peer review] (Case 2). Both were publicly funded, not purpose-built for stroke rehabilitation, with similar ward size and number of beds, but the buildings were of different ages (see S1 Table and Figs 1 and 2). Qualitative and quantitative data were collected in parallel from the two cases, analysed separately, and then merged in a convergent mixed-methods analysis [28, 29]. The two cases were compared throughout in a cross-case comparison. Hospital Research Ethics Approval was obtained (HREC/18/SVHM/210). Study design and reporting was informed by the consolidated criteria for reporting qualitative research [COREQ; 31], the standards for reporting qualitative research [SRQR; 32], and guidelines for reporting mixed methods research [33, 34].

## Participants

Stroke survivors admitted to the rehabilitation ward at either case were invited to participate during their stay. To promote equitable research and ensure representative findings [35], eligibility criteria were broad (see Table 1). Input from aphasia experts ensured participant-facing materials were accessible (large fonts, active voice, everyday words, pictures, etc.). Data collection procedures were piloted with a stroke survivor consumer advisor.

Nonprobabilistic convenience sampling was used. We intended to recruit 20 participants at each case, but the final sample size was based on saturation of themes from analysis of the qualitative data [36].

## Data collection

Checklists were completed to describe the physical environment of the buildings (S1 File). Photographs, floor plans, and descriptive and reflective field notes taken throughout data collection helped to further describe the physical environment and interactions between the environment and people at each case [37]. Clinical and demographic data were collected to

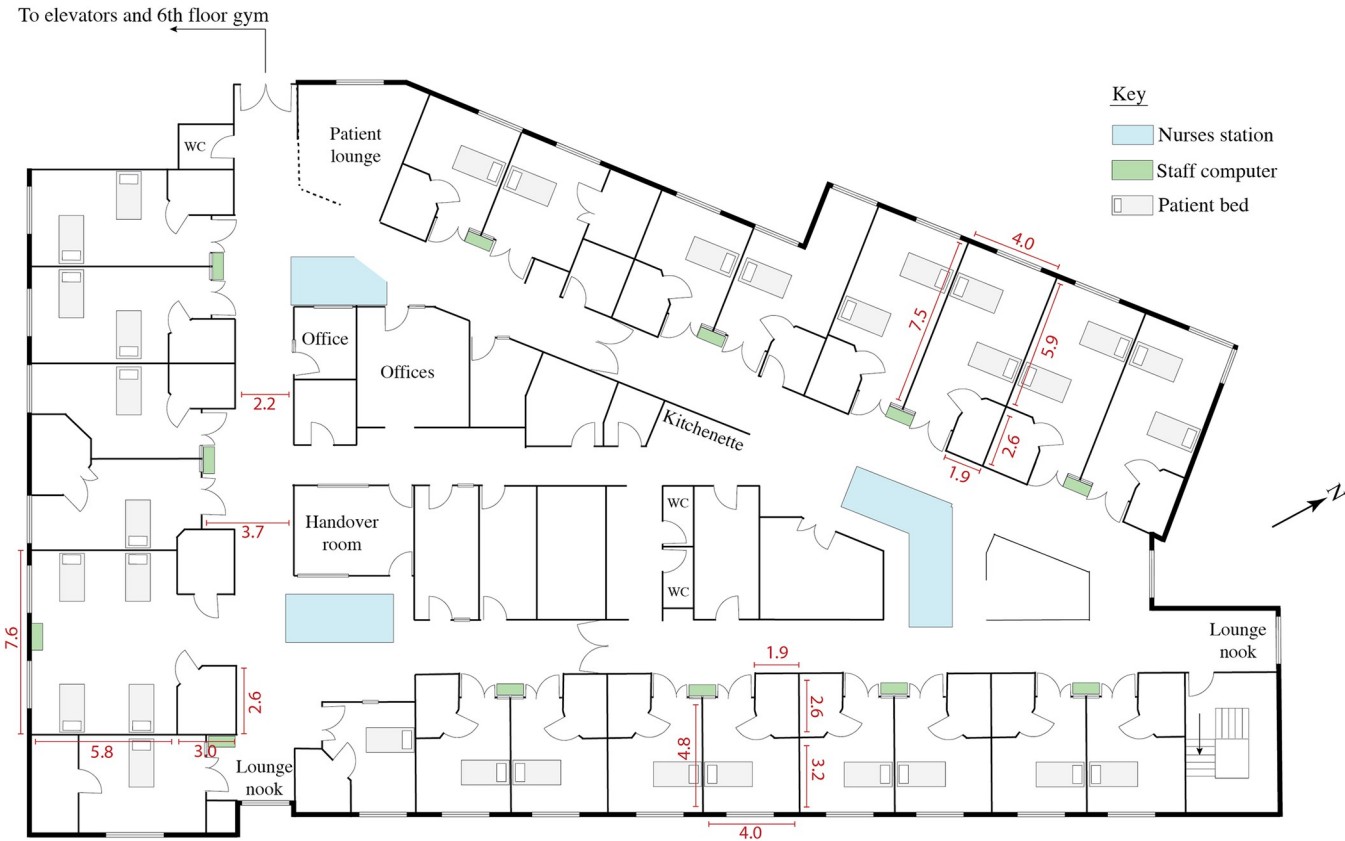

**Fig 2. Floor plan of the inpatient rehabilitation ward at Case 2.** The ward and gym are located on the 6[th] floor. All measurements are in meters. Plan is not to scale.

describe participant characteristics, following standard recommendations [38]. Author [INITALS] collected all data between September 2018 and June 2019.

**Qualitative data collection.** All participants completed a one-on-one walk-through semi-structured interview, following a route of their choosing through the facility [39, 40]. Interview questions were designed to explore stroke survivor experience of the physical environment, and their behaviour (activity and rest), emotional well-being, and safety in this setting

**Table 1. Participant eligibility criteria.**

| Criteria |
| --- |
| Inclusion |
| 1. ≥ 18 years of age<br>2. Current rehabilitation inpatient at one of the two case facilities<br>3. Primary reason for admission was for rehabilitation after a stroke (any type of stroke)<br>4. Admitted to the rehabilitation ward at least 3 days prior to commencing participation, and expected to remain an inpatient for the duration of their study participation |
| Exclusion |
| 1. Severe cognitive or language impairments (e.g., global aphasia or severe comprehension deficits) to an extent that would preclude participation as identified by a speech pathologist or other member of the clinical team. Patients with less severe forms of cognitive or language impairments were eligible to participate if they or a proxy could provide informed consent.<br>2. Non-English speaking (i.e., require an interpreter) |

(S2 File). Participants with aphasia could draw or take photographs instead of verbalising, and those who found it prohibitive to mobilise could choose to remain in-situ using photographs of the ward to prompt discussion. Interviews were audio-recorded.

**Quantitative data collection.** To explore stroke survivor behaviour in the environment, data were collected about each participants' activity (physical, cognitive, and social) and sleep. Behavioural mapping was used to observe their activity [41]. Observations were made for five seconds every 10 minutes over a nine-hour period on any weekday, generating 54 discrete observation epochs per participant including three randomly allocated observer breaks. At each epoch, the observer recorded the participant's physical, cognitive, and social activity, location, and who they were with. At each epoch, the participant was presumed to have participated in that activity for the preceding ten minutes. Physical activity included any purposeful physical movement. Cognitive activity was defined as being engaged in a cognitive task (e.g., reading, puzzles, listening to music), and social activity included any verbal or non-verbal interaction. If the participant could not be observed (e.g., observer break, participant in the bathroom), staff or the participant were asked to retrospectively estimate activity.

Participants completed a questionnaire booklet comprising the Pittsburgh Sleep Quality Index (PSQI) questionnaire [42] –adapted to refer to the past week rather than the past month to correspond more closely with the participant's inpatient stay–and a suite of emotional well-being questionnaires: the Depression, Anxiety, and Stress Scales [DASS-21; 43]; the Multidimensional State Boredom Scale [MSBS; 44]; and a visual analogue scale (VAS) designed to assess the extent to which participants thought the hospital environment motivated them to participate in rehabilitation practice and activity from zero (strongly demotivating) to 100 (strongly motivating). Participants could choose to complete the booklet independently or with help, at any point in their participation, and over as many sessions as needed.

To address patient safety, records of reported stroke patient falls were obtained retrospectively for the 12-month period preceding data collection at each case. The falls records included a patient identifier, date of fall, any injuries resulting from the fall, and a brief description of how and where the fall occurred.

## Analysis

**Qualitative data analysis.** Interviews were transcribed verbatim and uploaded with associated field notes and photographs to QSR International's NVivo 12 software [45]. Qualitative data were analysed inductively using generic qualitative inquiry, which is a particularly suitable approach for exploratory research [46]. Triangulation was sought between textual and pictorial data, between participants, and between cases. After reviewing the interview data from both cases, four particularly long and descriptively 'rich' interviews (two from each case) were independently coded by both [INITIALS] and [INITIALS]. Open, or emergent, coding was used to label units of information [47]. The coders then met to develop consensus codes which [INITIALS] built upon while analysing the remainder of the interviews. Initial coding was completed for Case 1 before Case 2 to allow for comparison between cases. Similar codes were then iteratively combined into larger categories of information, called themes, using axial coding [36]. The themes were defined for both cases concurrently allowing for reflection back and forth between the cases. For member checking, a results summary was sent in December 2019 to all participants who had consented to be contacted (S3 File). To further ensure reliability and validity, a peer review workshop was held in September 2019 where [INITIALS of authors], a stroke survivor consumer advisor, a healthcare environments expert, and a qualitative analysis expert reviewed all codes and achieved consensus on the final themes.

**Quantitative data analysis.** All quantitative data were managed using the Research Electronic Data Capture (REDCap) platform [48], analysed and visualised using R software for statistical computing [49]. Descriptive statistics (including counts and percentages; means and standard deviations; or medians and interquartile ranges, as appropriate) were used to summarise participant demographics, questionnaire responses, and the retrospective falls records. Questionnaires not completed in full were considered missing data and excluded from analysis. For the behavioural mapping data, descriptive statistics (median and interquartile range) were used to report activity, people present, and location of all participants and on average for both cases. Counts of each physical, cognitive, and social activity, number and type of people present, and location category across the day were calculated as percentages of the total number of observations per participant. This provided an estimate of the amount of time that a participant spent undertaking each type of activity, or combination of activities, as a proportion of the participant's total observation time. Unobserved epochs where behaviour could not be estimated were excluded from analysis unless they could be considered missing at random (e.g., observer break).

In keeping with the exploratory approach of this study, comparative statistics were not conducted. Instead, results for each case were compared to population norms where possible using $z$ scores, and data were graphed to visually show relationships between aspects of the physical environment and patients' behaviour and emotional well-being within and between cases.

**Convergent mixed-methods analysis.** Joint display tables were used to make a side-by-side comparison of the qualitative and quantitative findings and merge (or converge) these findings [28, 29]. Four joint display tables were completed, one for each of the constructs central to this case study: activity, sleep, emotional well-being, and safety. The contents within each of the joint display tables were organised according to 'topics' drawn from the important aspects of the qualitative themes and the key findings from the quantitative analysis, and which were agreed upon by the attendees of the peer review workshop (see S1 Fig). Adjacent qualitative and quantitative findings were described as either 'congruent' (agreeing with each other), 'divergent' (differing from each other), or 'unique' (only qualitative or only quantitative findings available). The results of the joint display tables–including the convergent, divergent, and unique findings–were then summarised in a narrative integration which provided an overarching explanation of the converged findings. This integration is expressed as a conceptual model of the role of the physical environment of inpatient rehabilitation facilities in patients' behaviour, emotional well-being, and safety after stroke, thereby addressing the aim of this study.

## Results

Twenty patients participated at Case 1 (45% female, mean age = 73 years) and 16 at Case 2 (37.5% female, mean age = 67.2 years). Approach, consent, and data completion rates are shown in Fig 3. No participants withdrew, but some questionnaire data were missing due to one Case 2 participant with aphasia choosing not to complete any questionnaires and eight participants (five from Case 1, two from Case 2) accidentally leaving question/s blank. Participant demographics, clinical information, and allocated bedroom type (single or multi-bed room) are summarised in Tables 2–4.

### Qualitative findings

All participants completed an interview. Interview duration averaged 47 minutes at Case 1 (range = 10–82 minutes) and 30 minutes at Case 2 (range 12–58 minutes). Analysis revealed

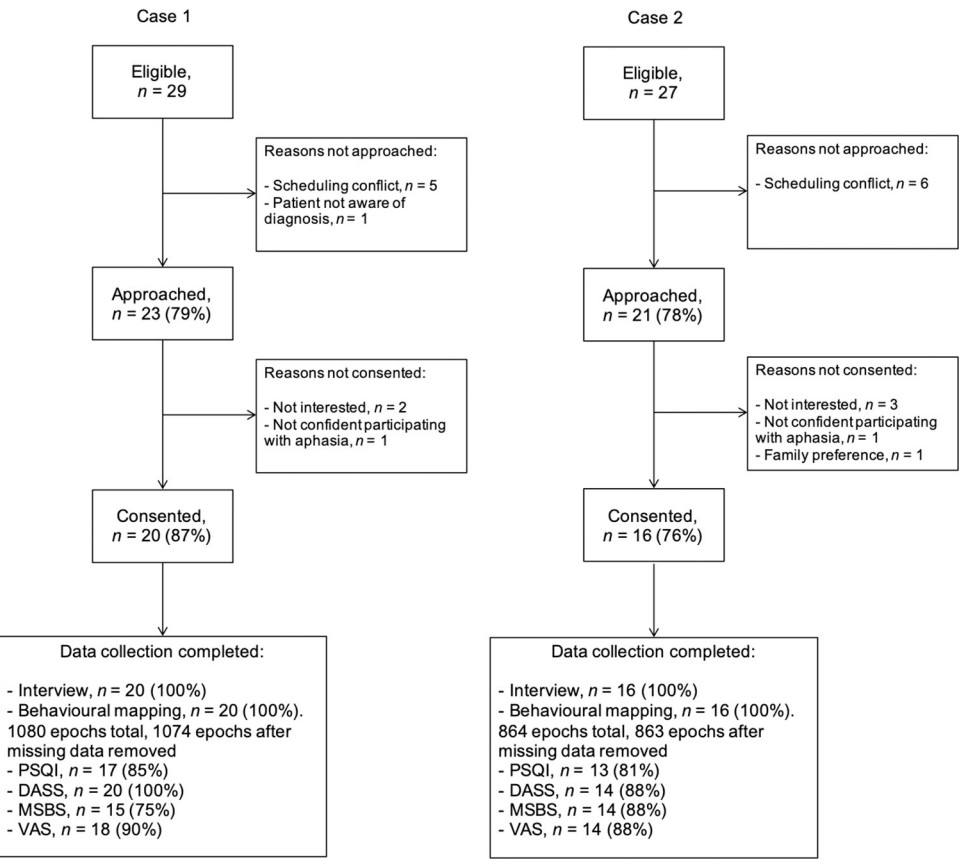

**Fig 3. Approach, consent, and data completion rates for both cases.**

four themes and 11 sub-themes, summarised in Table 5 (see S2 Table for further illustrative quotes). Further description and cross-case comparison are detailed in full elsewhere [50].

## Quantitative results

**Activity.** Behavioural mapping was completed for all participants (see Fig 3). Fig 4 shows the time spent in physical, cognitive, and social activity in each location at each case. Participants at both cases, and especially Case 2, were inactive for most of the time that they were in their bedrooms. Most activity occurred outside the bedroom, particularly in bathrooms, communal areas, therapy areas, and hallways, but participants spent very little time at these locations. Most participants spent over 70% of observations in their bedroom: median 75% at Case 1 (IQR 69.9%, 81.5%) and 79.6% at Case 2 (IQR 70.8%, 87.5%). Participants visited a median of four different locations during the day, including their bedroom (IQR 3, 4.3 at Case 1; IQR 3.8, 5 at Case 2). After bedrooms and therapy spaces, hallways were the next most frequently visited location. One participant at Case 2 spent over 60% of the day in the hallway sitting in one of the lounge nooks. Participants spent no time outdoors at Case 2, very little time outdoors at Case 1 (mean = 0.3%; median = 0%, IQR 0%, 0%), and very little time in the communal areas at either case (mean = 2.2%, median = 0%, IQR 0%, 2.4% at Case 1; mean = 0.9%, median = 0%, IQR 0%, 0% at Case 2).

Participants at Case 1 were alone for a median of 38% of observations (IQR 16.7%, 59.3%) and participants at Case 2 were alone for 34.3% of observations (IQR 19.3%, 37.5%).

**Table 2. Demographics of the participants at each case.**

| Demographic information | Case 1 *n* = 20 | Case 2 *n* = 16 |
|---|---|---|
| Age, mean (range) | 73.0 (61–90) | 67.2 (43–89) |
| Sex, *n* (%) | | |
| Female | 9 (45.0) | 6 (37.5) |
| Male | 11 (55.0) | 10 (62.5) |
| First language is English, *n* (%) | 16 (80.0) | 15 (93.8) |
| Employment pre-stroke, *n* (%) | | |
| Full-time | 2 (10.0) | 4 (25.0) |
| Part-time | 3 (15.0) | 1 (6.3) |
| Not working | 15 (75.0) | 11 (68.7) |
| Education, *n* (%) | | |
| No education | 0 | 1 (6.3) |
| Primary school | 7 (35.0) | 8 (50.0) |
| Secondary school | 4 (20.0) | 3 (18.7) |
| Diploma TAFE or apprentice | 4 (20.0) | 4 (25.0) |
| Undergraduate university | 4 (20.0) | 0 (0.0) |
| Postgraduate university | 1 (5.0) | 0 (0.0) |
| Accommodation pre-stroke, *n* (%) | | |
| Own home alone | 11 (55.0) | 3 (18.7) |
| Own home with others | 9 (45.0) | 12 (75.0) |
| Residential aged care | 0 (0.0) | 0 (0.0) |
| Retirement village | 0 (0.0) | 1 (6.3) |

Participants were completely inactive for a median of almost a quarter of the nine-hour observation period at Case 1 (median = 22.2% [2 hours], IQR 12.9%, 37.5%) and a third at Case 2 (median = 34.3% [3.1 hours], IQR 18.1%, 14.3%). Overall, participants at Case 1 appeared to spend more time in physical activity than participants at Case 2 (median = 50%, IQR 38%, 67.6% for Case 1; median = 37%, IQR 27.3%, 44% for Case 2), and more time in cognitive activity (median = 14.8%, IQR 8.33%, 26.4% for Case 1; median = 2.8%, IQR 0%, 13.4% for Case 2), but the proportion of time spent in social activity was similar at both cases (median = 40.7%, IQR 23.6%, 48.6% for Case 1; median = 30.6%, IQR 25.9%, 43.1% for Case 2). Participants at Case 1 appeared to spend more time in physical and cognitive activity while in therapy areas compared to Case 2 (see Fig 4). There were a similar number of other people in the therapy areas at both cases, but the gym was smaller, and therefore more crowded, at Case 2 (see S1 Table). Findings regarding activity and time spent alone in single and shared bedrooms and for participants with language and/or cognitive impairments are included in S4 File.

**Sleep.** Patient-reported sleep quality was similarly poor at both cases and participants in single-bed rooms reported slightly worse sleep than participants in shared bedrooms (see Table 6).

**Emotional well-being.** According to DASS recommended cut-off scores [51], more participants at Case 2 experienced depression, anxiety, and/or stress outside the normal range than at Case 1 (30% at Case 1 and 81.3% at Case 2 experienced depression; 30% at Case 1 and 68.7% at Case 2 experienced anxiety; 30% at Case 1 and 68.7% at Case 2 experienced stress)–see S4 File for median scores. At both cases, the mean boredom scores were within half a standard deviation of population norms (i.e., *z* score ≤ 0.5), except for the scores on the time perception subscale, which appeared higher than population norms, see S4 File. This subscale

**Table 3. Clinical information of the participants at each case.**

| Clinical information | Case 1 *n* = 20 | Case 2 *n* = 16 |
|---|---|---|
| Days since stroke[a], mean (*SD*, range) | 36.2 (25.5, 9–116) | 34.3 (27.7, 5–114) |
| Days in rehab at participation, mean (*SD*, range) | 18.2 (14.3, 4–66) | 19.6 (17.8, 3–66) |
| First stroke, *n* (%) | 13 (65) | 10 (62.5) |
| **Stroke type, *n* (%)** | | |
| Ischaemic | 15 (75) | 13 (81.3) |
| Haemorrhagic | 5 (25) | 3 (18.7) |
| **Pre-stroke walking, *n* (%)** | | |
| Walking independently no aids | 16 (80) | 14 (87.5) |
| Walking with gait aid | 4 (20) | 2 (12.5) |
| **No. of comorbidities, *n* (%)** | | |
| None | 4 (20) | 5 (31.3) |
| One | 12 (60) | 10 (62.5) |
| Two | 3 (15) | 1 (6.3) |
| Three | 1 (5) | 0 (0) |
| **Type of comorbidity, *n* (%)[b]** | | |
| Cognitive decline | 2 (10) | 1 (6.3) |
| Osteoarthritis | 6 (30) | 0 (0) |
| Other | 13 (65) | 11 (68.7) |
| **Pre-stroke mRS, *n* (%)** | | |
| No symptoms | 7 (35) | 5 (31.3) |
| No disability despite symptoms | 11 (55) | 7 (43.7) |
| Slight disability | 2 (10) | 1 (6.3) |
| Moderate disability | 0 (0) | 3 (18.7) |
| Moderately severe disability | 0 (0) | 0 (0) |
| Severe disability | 0 (0) | 0 (0) |
| **Stroke severity, NIHSS, median (IQR)** | 5 (6) | 7 (5.5) |
| Minor (1–4), *n* (%) | 9 (45) | 4 (25) |
| Moderate (5–15), *n* (%) | 11 (55) | 11 (68.7) |
| Moderate to severe (16–20), *n* (%) | 0 (0) | 1 (6.3) |
| Severe (21–42), *n* (%) | 0 (0) | 0 (0) |
| **Aphasia or dysarthria, *n* (%)[c]** | 7 (35) | 7 (43.7) |
| **MoCA, median (IQR) [d, e]** | 21.5 (4.5) | 19 (6) |
| Normal, MoCA ≥26, *n* (%) | 4 (20) | 1 (6.3) |
| Mild, MoCA 18–25, *n* (%) | 13 (65) | 7 (43.7) |
| Moderate, MoCA 10–17, *n* (%) | 1 (5) | 5 (31.3) |
| Severe, MoCA <10, *n* (%) | 2 (10) | 0 (0) |
| Unknown, MoCA incomplete | 0 (0) | 3 (18.7) |
| **Mobility at participation, *n* (%)[f]** | | |
| Independently mobile | 12 (60) | 7 (43.7) |
| Dependent for mobilisation | 8 (40) | 9 (56.3) |
| **FIM admission, mean (*SD*, range)** | 66.8 (25, 23–101) | 53.1 (19.7, 23–84) |
| **FIM discharge, mean (*SD*, range)** | 95 (27.0, 48–126) | 80.6 (26.6, 27–117) |
| **FIM difference, mean (*SD*, range)** | 28.2 (15.2, 2–60) | 27.5 (15.3, 3–56) |
| **Length of stay in rehab (days), mean (*SD*, range)** | 42.2 (27.9, 8–108) | 45.6 (30.5, 14–127) |
| **Discharge destination, *n* (%)** | | |
| House or flat—alone | 3 (15) | 1 (6.3) |
| House or flat—with others | 10 (50) | 5 (31.3) |

(*Continued*)

**Table 3.** (Continued)

| Clinical information | Case 1 *n* = 20 | Case 2 *n* = 16 |
|---|---|---|
| Residential aged care | 2 (10) | 0 (0) |
| Retirement village | 1 (5) | 0 (0) |
| Transition care planning | 1 (5) | 5 (31.3) |
| Other hospital ward | 3 (15) | 5 (31.3) |

Acronyms: FIM = Functional Independence Measure, mRS = modified Rankin Score, NIHSS = National Institute of Health Stroke Scale, MoCA = Montreal cognitive assessment, *SD* = standard deviation.

[a]At start of participation in the study

[b]The *n* is larger than the total number of participants with comorbidities because some participants had more than one comorbidity.

[c]The presence of aphasia and dysarthria determined by the NIHSS. Scores of 1, 2, or 3 on question 9 of the NIHSS indicate aphasia. Scores of 1 or 2 on question 10 of the NHISS indicate dysarthria.

[d]The MoCA was not completed by 3 participants at Case 2: one had moderate-severe aphasia (score of 2 on NIHSS question 9, 'best language'); one was not sufficiently fluent in English and had dysarthria (MoCA was attempted with a speech pathologist but was untenable); and one had pervasive loss of vision (score of 2 on NIHSS question 3, 'visual', indicating complete hemianopia).

[e]The severity levels are recommended by MoCA but have not yet been established by research.

[f]Patients who could mobilise independently without aids or using a walking frame or self-propelled wheelchair without the help of another person are classified as independently mobile. Patients who require another person to assist with their locomotion are classified as dependent for mobilisation.

measures the extent to which respondents feel that time is distorted and/or moving slowly [44]. Regarding motivation, the median score on the VAS was 63 at Case 1 (IQR 45.5, 73.5) and 40 at Case 2 (IQR 7, 84). Further results describing emotional well-being in single and shared bedrooms are reported in S4 File.

**Safety.** Retrospective falls records indicated that 25% of stroke patients experienced a fall at Case 1 and 21% at Case 2 during the 12-month audit period. Table 7 shows the location, context, and outcome for reported falls. Three of the Case 1 falls resulted in minor injury (13.7%) including grazed knee, hip pain, and skin tear. All fall injuries at Case 1 occurred in the bedroom; two because of falling from bed (one while trying to reach eyeglasses) and one due to tripping over an object while reaching for a mobile phone. Seven of the Case 2 falls

**Table 4. Participant bedroom type for both cases.**

| Bedroom type[a], *n* (%) | Case 1 *n* = 20 | Case 2 *n* = 16 |
|---|---|---|
| **Single-bed room, any type** | 13 (65) | 6 (37.5) |
| Single, ensuite | 9 (45) | 6 (37.5) |
| Single, shared bathroom | 1 (5) | 0 (0) |
| Single, kitchen, shared bathroom | 3 (15) | 0 (0) |
| **Shared bedroom, any type** | 7 (35) | 10 (62.5) |
| Two-bed room | 0 (0) | 7 (43.7) |
| Three-bed room | 4 (20) | 0 (0) |
| Four-bed room | 3 (15) | 3 (18.8) |

[a]Room type was recorded at the time of behavioural mapping. One participant in Case 1 changed rooms after behavioural mapping and before the interview from a four-bed room to a single-bed room with kitchen and shared bathroom.

**Table 5. The themes and sub-themes at both cases and illustrative quotes.**

| Theme | Sub-theme | Description, cross-case comparison, and quotes |
|---|---|---|
| **1. Entrapment and escape** | 1.1. The environment is restrictive and boring. | Spending too long in one room or too much time alone contributed to this feeling–*"[My bedroom] stinks . . . because I'm sitting here all day . . . It's not good for the brain . . . it's still locking you up"* (2656–08)–and prompted some participants to try and leave the ward. Despite the larger bedrooms, the sense of entrapment persisted for at least half of the participants at Case 2, independent of their type of impairment or whether they were in a single or shared bedroom —*"just seeing people over and over and over and over and hearing the same things over and over and over and over oh. . . . where else is there to bloody go!? Nowhere, is there? That's what I mean. You're too enclosed in the place."* (2657–15) |
| | 1.2. Connection, purpose, and change are means of escape | Purposeful activities and objects (e.g., therapy equipment) provided physical markers of recovery and progress. Participants appreciated opportunities to leave the ward, e.g., therapy sessions in the gym or going to the café: *"[the café] doesn't make you feel like you're in hospital . . . you see people come and go and you know yeah sort of sort of feeling normal . . . feeling of normalcy"* (2657–04). Trips outside the hospital building were especially appreciated. Connection to other people and to the outside world, views from a window, and variety and change in the environment, helped participants to feel tethered to *"the world going about its business"* (2656–16). *"I [think] the windows [are] good . . . had a bit of a view . . . you don't feel so isolated I suppose, feel more a part of things"* (2657–04). *"[looking out the window] sorts of fills your mind. Occupies your mind . . . keeps your mind active."* (2657–12). |
| | 1.3. Nature and the outdoors are means of escape | Nature and the outdoors were an important sensory experience and provide relief from feelings of restriction or boredom: *"much better than sitting in that dark [bed]room. . . . The fresh air. The green grass. . . . The open air that's what I like. Freedom"* (2656–08). |
| **2. Power, dependency, and identity in an institutional environment** | 2.1. Surveillance | Participants felt that staff, policies, procedures, and surveillance tools such as bed alarms held power over their movements and activities, akin to being *"in a prison"* (2656–11), *"in school"* (2656–12), or *"in the colosseum"* (2656–01). *"What's wrong with me going outside the ward by myself though? . . . I shouldn't have to ask!"* (2657–15). Surveillance tools prompted some participants to consider sneaking out or self-discharge, and also interrupted sleep. |
| | 2.2. Relying on the environment for safety and well-being | Connection with staff through proximity to the nurses' station and access to the nurse call-bell was an important safety net–*"if I don't [feel safe then] I know this [call-bell] is here. . . . without that I feel I'm cut off"* (2656–17). Visual and aural aesthetics played an important role in well-being, with the *"beige"* (2656–01) ward at Case 1 providing no reason to leave the bedroom–*"there's nothing interesting out there"* (2656–09)–and the new ward at Case 2 feeling *"reassuring . . . because it's modern you feel safer in it . . . it's got everything in it that you need"* (2657–02). |
| | 2.3. Ownership of the environment | Physical barriers such as locked doors indicated that the ward belonged to staff, not patients. Bedrooms were difficult to personalise, especially at Case 2 where policy dictated that nothing be stuck on the walls of the new building. Some participants did not feel that they belonged in the space, nor that the space belonged to them. Others felt a sense of belonging, especially in the gym since it contained equipment that was specifically for them. Being unable to accommodate visitors (e.g., lack of space or shortage of chairs) was a frustrating reminder of participants' lack of control over the environment–*"those people have come to see me and if I can't accommodate them to a point where they're comfortable . . . then it defeats the purpose of the exercise of the visit"* (2656–13). |
| **3. The rehabilitation facility is a shared space** | 3.1. Compromise, privacy, and peer support in the bedrooms | Bedrooms were a site of compromise, with noise levels and privacy a concern in both shared and single rooms. Noise from inside and outside the bedroom impacted sleep. Participants wanted privacy but did not want to be isolated. Peer support occurred in shared bedrooms and some participants chose to remain in a shared bedroom even when a single one was offered–*"I'm very happy with the ladies [in my room]. We can talk during the night . . . we always try to find ways to come close."* (2656–06, Case 1, four-bed room). |
| | 3.2. Social connection in hallways and therapy spaces | Verbal exchange was not always essential; just seeing other stroke survivors–*"people like myself"* (2656–07)–in hallways and therapy areas was enough to promote feelings of connection, comradery, and empathy. Although sometimes upsetting to see other people in distress, seeing others improve in therapy gave a sense of proportion and hope. The large dimensions of the gym at Case 1 made it hard for some to feel part of a community–*"it's sort of too big and that for me to join anything"* (2656–20). |
| | 3.3. Seeking defined communal spaces | Many participants desired designated communal spaces, as opposed to hallways and therapy spaces in which social connection was only incidental. There were participants at both cases who did not know the lounge existed, found it awkwardly shaped or hard to find (Case 1), and devoid of people–*"there's hardly ever anybody [there]. I'm gonna to sit there on my own? I might as well come [to my bedroom] and sit in here. Yeah so I don't use that much."* (2656–04). The hospital café was preferred and often utilised by participants and their visitors, but was not open on weekends. |

*(Continued)*

**Table 5.** (Continued)

| Theme | Sub-theme | Description, cross-case comparison, and quotes |
|---|---|---|
| **4. The environment should be legible and patient-centred** | 4.1. (Dis)orientation in the physical environment | Many participants had no knowledge of the layout of the building–*"[I] didn't even know how many floors there were"* (2656–06)–were ignorant of the location or existence of key spaces and were easily disoriented–*"all looks the same"* (2657–10). They wanted to know what was available, where, the purpose of these spaces, who could use them, and any access restrictions (e.g., opening hours). Lacking this knowledge, they stayed in their bedroom. |
| | 4.2. Convenience and suitability of the environment (fit-for-purpose) | Participants noticed aspects of the environment that were helpful for their care (e.g., therapy equipment, wide hallways for wheelchairs, nurse call bell) and those that were inefficient, confusing, did not suit their needs, or actively hindered their independence (e.g., power points or call bell out of reach, doorway to balcony too narrow for wheelchair). Lack of maintenance or attention led some environmental features to become antithetical to their intended purpose, which made some participants feel that the environment had set them up to fail–*"My bed [at home is] big. Look how little these beds are! Is it any wonder [I] fell out?"* (2656–08). In another example, a timetable–the aim of which is to help participants prepare for therapy and feel in control of their time–was placed too far away for the participant to read and so became another reminder of his dependency: *"I do [like having the timetable] . . . I get cross because I can't read it without my glasses. And I can't reach my glasses easily."* (2657–09) |

resulted in injury (14.6%), but the nature of the injury was only reported for two of these (skin tear and hit head). Two of the seven falls occurred in the bedroom (both a fall from bed), four in the bathroom (one fall from chair, one fall while standing/walking, and two reason unrecorded), and one location not recorded. Of the 22 falls at Case 1, four (18%) were brought to the attention of staff by a proximate alarm, but the patients had already fallen when staff

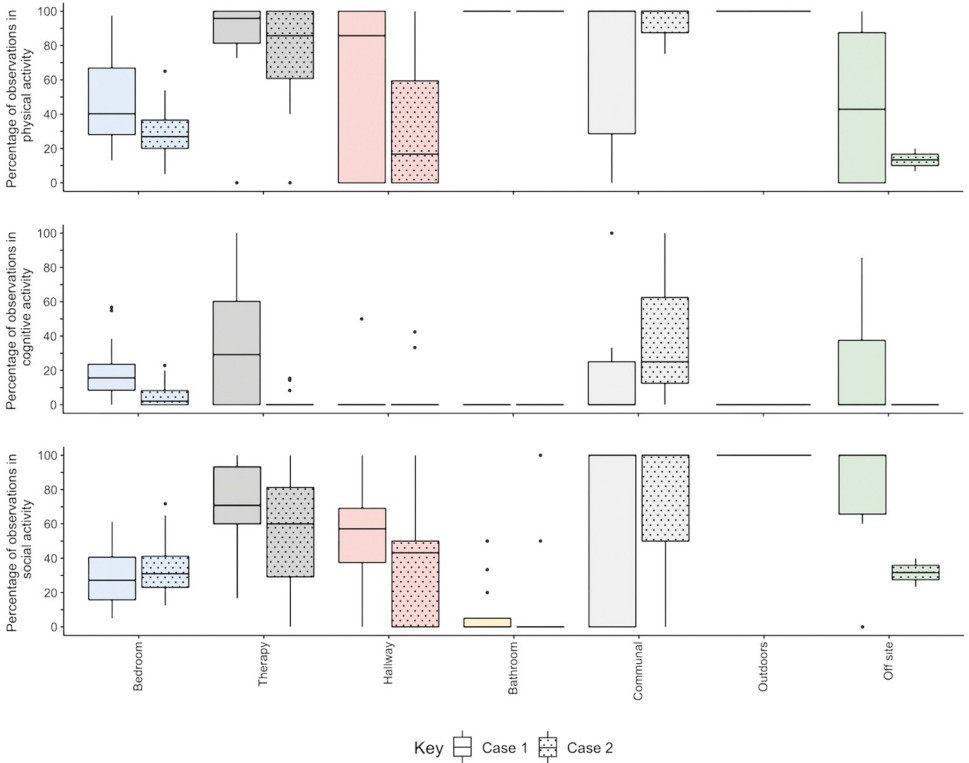

**Fig 4. Median percentage of observations in physical, cognitive, and social activity in each location at each case.** The number of observations that participants spent in physical, cognitive, and social activity as a percentage of total observations in each location. Boxes indicate median and interquartile range. Dots represent outliers.

**Table 6. Patient-reported sleep quality in single and shared bedrooms.**

| PSQI | Case 1 *n* = 20 | Case 2 *n* = 16 |
|---|---|---|
| PSQI completed | *n* = 17[a] | *n* = 13[a] |
| PSQI, median (IQR, range) | 10 (5, 1–15) | 10 (6, 2–15) |
| Participants in single | 10 (5, 1–15), *n* = 11 | 11 (3, 8–14), *n* = 5 |
| Participants in shared | 9 (7.75, 3–15), *n* = 6 | 7.5 (8.75, 2–15), *n* = 8 |

Single = single-bed room, shared = shared bedroom. Acronyms: PSQI = Pittsburgh Sleep Quality Index, IQR = interquartile range. The PSQI has a minimum score of 0 and a maximum score of 21. Lower PSQI scores indicate better sleep quality and any score above 5 is associated with poor sleep quality [42].
[a]PSQI data missing due to one participant (Case 2) with aphasia choosing not to complete any questionnaires and five (three from Case 1, two from case 2) accidentally leaving question/s blank.

arrived. A further two patients at Case 1 had been assigned a proximate alarm prior to their fall, but one had failed to go off and the other couldn't be located after the fall. A sensor mat which had failed to alarm was mentioned in relation to one fall at Case 2.

## Mixed-methods results and narrative integration

The contents of the joint display tables (Tables 8–11) are organised according to seven topics (see S1 Fig): 1) Extended time spent in one location (namely, the bedroom); 2) The presence and absence of other people (in the bedrooms, hallways, and communal spaces); 3)

**Table 7. Location, context, and outcome of reported falls for stroke patients.**

| Location, context, and outcome of falls | Case 1 *n* = 22 | Case 2 *n* = 48 |
|---|---|---|
| Location, *n* (%) | | |
| Bedroom | 19 (86.4) | 20 (41.7) |
| Bathroom | 3 (13.6) | 18 (37.5) |
| Not reported | 0 (0) | 10 (20.8) |
| Staff present, *n* (%) | | |
| Yes | 9 (40.9) | 4 (8.3) |
| No | 13 (59.1) | 34 (70.8) |
| Not reported | 0 (0) | 10 (20.8) |
| Reason for fall[a], *n* (%) | | |
| Fall from bed | 9 (40.9) | 16 (33.3) |
| Transfer (assisted) | 4 (18.2) | 4 (8.3) |
| Reaching for object | 3 (13.6) | None reported |
| Trip over obstacle | 3 (13.6) | None reported |
| Transfer (unassisted) | 2 (9.1) | 1 (2.1) |
| Fall from chair | 2 (9.1) | 9 (18.8) |
| Fall while standing or walking | 2 (9.1) | 8 (16.7) |
| Unknown or not reported | 1 (4.6) | 10 (20.8) |
| Injury reported, *n* (%) | | |
| No | 19 (86.4) | 41 (85.4) |
| Yes | 3 (13.7) | 7 (14.6) |

[a]At Case 1, one fall occurred because of a combination between reaching for an object, standing, and tripping over an obstacle, and two falls occurred because of a combination of reaching for an object and falling from bed, so the total of 'Reasons for fall' = 26 at Case 1.

**Table 8. Comparison of the qualitative and quantitative findings that relate to patients' physical, cognitive, and social activity.**

| Topic | Qualitative findings | Quantitative findings | Convergence |
|---|---|---|---|
| **Extended time spent in one location** | **Theme 1: Entrapment and escape**<br>• Participants felt they spent too long in their bedroom (sub-theme 1.1).<br>• *Going out of the bedroom and ward encourages more purposeful and beneficial activity (sub-themes 1.2 and 1.3).*<br>• Some participants sought out activity, most responded with inactivity (sub-theme 1.1).<br>• *Participants desired more connection to the outside world (sub-themes 1.2 and 1.3).* | **Behavioural mapping**<br>• Participants spent majority of day in bedroom and visited few locations.<br>• *Participants more active in other locations than in bedroom (Fig 4).*<br>• Extent of activity varied widely between participants (IQRs in Fig 4).<br>• *Participants at Case 1 rarely went outside and at Case 2 never went outside (Fig 4).* | **Congruent**<br>• Extended periods in bedroom.<br>• *More active outside of bedroom.*<br>• Some patients seek activity, some do not.<br>**Divergent**<br>• *Patients want to go outside, but do not go outside.* |
| **Presence and absence of other people–bedrooms** | **Theme 1: Entrapment and escape**<br>• Participants felt they spent too much time alone in their bedroom (sub-theme 1.1).<br>**Theme 3: Rehab facility is a shared space**<br>• *Shared bedrooms provide opportunity to be with others but only lead to social interaction/support if roommates are able/well enough to interact (sub-theme 3.1).* | **Behavioural mapping**<br>• Participants were alone for ~1/3 of the day.<br>• *Participants in single-bed rooms spend more time alone, especially if had language or cognitive impairments (S4 File).*<br>• Time spent with others more 'productive' (i.e., more social activity) in single- than shared bedrooms (S4 File).<br>*Many variables may influence how much time is spent alone, inc. roommates' impairments, position of beds, barrier between beds, and location of room (S4 File).* | **Congruent**<br>• Often alone in bedroom.<br>• *Less alone in shared bedrooms, but not more socially active.*<br>• *More alone if you or roommate have language and/or cognitive impairments.*<br>**Quantitative only**<br>• Being with others may result in more social activity in single-bed room than in shared.<br>• *Other variables besides whether in single or shared bedroom impact how much time is spent alone in bedroom.* |
| **Presence and absence of other people–hallways** | **Theme 3: Rehab facility is a shared space**<br>• Hallways provide opportunity for social activity (sub-theme 3.2).<br>• *The lounge nooks at Case 2 were used as a place to be private, but not alone (sub-theme 3.2).* | **Behavioural mapping**<br>• Social activity in hallways (Fig 4).<br>• *Hallways used differently at Case 2; time spent in lounge nooks.* | **Congruent**<br>• Hallways provide spaces for social activity.<br>• *Lounge nooks provide spaces to be private but not alone.* |
| **Presence and absence of other people–communal spaces** | **Theme 3: Rehab facility is a shared space**<br>• Participants want designated communal spaces (sub-theme 3.3).<br>• *Some do not go to lounge if no one else is there, others do not go if lounge is too crowded (sub-theme 3.3).* | **Behavioural mapping**<br>• Participants spend very little time in the communal spaces. | **Divergent**<br>• Desire for communal spaces but do not use those available.<br>**Qualitative only**<br>• *Number of people in communal spaces can encourage or discourage use.* |
| **Surveillance** | **Theme 2: Power, dependency, and identity in an institutional environment**<br>• Bed alarms (Case 1) and being watched (Case 2) perceived as punishments, so stay in bedroom and less physical activity (sub-theme 2.1). | – | **Qualitative only**<br>• Surveillance may discourage activity. |
| **Interest or fascination** | **Theme 1: Entrapment and escape**<br>• Some found environment uninteresting; no reason to leave bedroom or ward and so more inactive. At Case 2, some found bedrooms pleasant (sub-theme 1.1). | **Behavioural mapping**<br>• May spend more time in bedrooms that are pleasant to be in at Case 2, compared to Case 1. | **Congruent**<br>• Interest and pleasantness may contribute to time spent in bedroom. |
| **Equipment, facilities, and technology** | **Theme 2: Power, dependency, and identity in an institutional environment**<br>• Personalising bedroom helped to spark social activity at Case 1; harder to personalise shared bedrooms and all bedrooms at Case 2 (sub-theme 2.3).<br>• *Participants felt that physical barriers (e.g., locked doors, opening hours) indicated that they needed permission to leave their bedroom (sub-theme 2.3).*<br>**Theme 4: The environment should be legible and patient-centred**<br>• Good quality, available therapy spaces and equipment facilitate practice (sub-theme 4.2).<br>• *Availability and accessibility of infrastructure (e.g., power plugs) and technology (e.g., touch screens) impact activity (sub-theme 4.2).* | **Behavioural mapping**<br>• Appears to be more social activity in the bedrooms in Case 2 compared to Case 1 (Fig 4).<br>**Behavioural mapping**<br>• No dedicated Occupational Therapy spaces at Case 2 (S1 Table) and appears to be less cognitive activity at Case 2. | **Divergent**<br>• Personalising bedroom is not the only determinant of social activity.<br>**Qualitative only**<br>• *Physical barriers may discourage activity.*<br>**Congruent**<br>• Availability of therapy spaces may facilitate therapy.<br>**Qualitative only**<br>• *Non-patient-centred infrastructure may discourage activity.* |

*(Continued)*

**Table 8.** (Continued)

| Topic | Qualitative findings | Quantitative findings | Convergence |
|---|---|---|---|
| **Layout and orientation** | **Theme 1: Entrapment and escape**<br>• Ward above ground level makes it harder to go outside but gives nice views (sub-theme 1.3).<br>**Theme 3: Rehab facility is a shared space**<br>• *Communal spaces need to be positioned so patients know they are available (sub-theme 3.3).*<br>**Theme 4: The environment should be legible and patient-centred**<br>• Participants do not feel oriented to ward, so do not know where to go except bedroom (sub-theme 4.1).<br>• *Circular layout of ward and wide hallways help with physical activity (sub-theme 4.2).* | **Behavioural mapping**<br>• Ward at Case 1 was closer to ground (S1 Table) and patients appeared to spend slightly more time outside than at Case 2. | **Congruent**<br>• Distance to outdoor areas may impact whether they are used.<br>**Qualitative only**<br>• *Out-of-the-way location of spaces may play a role in whether they are used.*<br>• Orientation to ward impacts where spend time.<br>• *Layout of ward and hallways impacts physical activity.* |
| **Dimensions of the space** | **Theme 3: Rehab facility is a shared space**<br>• To facilitate social interaction, communal spaces need to be sizeable to accommodate enough people without being crowded (sub-theme 3.3).<br>**Theme 4: The environment should be legible and patient-centred**<br>• *Case 1: large gym facilitates practice. Case 2: small gym increases wait-times and makes it harder to practice in therapy (sub-theme 4.2).* | **Behavioural mapping**<br>• *Pattern of activity in the therapy areas appeared to differ between cases; more physical, cognitive, and social activity at Case 1.*<br>• *More crowded in therapy areas at Case 2 compared to Case 1.* | **Qualitative only**<br>• Size of communal spaces may impact use.<br>**Congruent**<br>• *Larger gym may facilitate therapy.* |

Font style (italicised or plain) shows which results link together across the qualitative, quantitative, and converged findings.

Findings were consistent across both cases, unless otherwise specified.

The following mark indicates no data: –

**Table 9. Comparison of the qualitative and quantitative findings that relate to patients' sleep.**

| Topic | Qualitative findings | Quantitative findings | Convergence |
|---|---|---|---|
| **Presence and absence of other people–bedrooms** | **Theme 3: Rehab facility is a shared space**<br>• Noise in shared bedrooms could be decreased if number of beds decreased (sub-theme 3.1).<br>• *Noise interrupts sleep in both single and shared bedrooms; noise comes from adjoining rooms, staff entering the room, or talking in hallways (sub-theme 3.1).* | **Sleep questionnaire**<br>• Participants in single-bed rooms report slightly worse sleep than participants in shared bedrooms (may be due to differences in clinical profile) (Table 6).<br>• Sleep was poor in both single and shared bedrooms (Table 6). | **Divergent**<br>• Patients feel that sleep could be improved in rooms with fewer beds, but quantitative results indicate that sleep is impacted in single-bed rooms (may be due to differences in clinical profile).<br>**Qualitative only**<br>• *Noise from outside room interrupts sleep in both single and shared bedrooms.*<br>**Quantitative only**<br>• Sleep is poor in both single and shared bedrooms. |
| **Surveillance** | **Theme 2: Power, dependency, and identity in an institutional environment**<br>• Participants at Case 1 commented that bed alarms interrupt sleep (sub-theme 2.1). | – | **Qualitative only**<br>• Bed alarms may interrupt sleep. |
| **Equipment, facilities, and technology** | **Theme 3: Rehab facility is a shared space**<br>• Noise comes from equipment, facilities, and technology (e.g., banging doors, alarms from call-bells) (sub-theme 3.1). | – | **Qualitative only**<br>• Ambient noise from equipment, facilities, and technology may interrupt sleep. |

Font style (italicised or regular) shows which results link together across the qualitative, quantitative, and converged findings.

Findings were consistent across both cases, unless otherwise specified.

The following mark indicates no data: –

**Table 10. Comparison of the qualitative and quantitative findings that relate to patients' emotional well-being.**

| Topics | Qualitative findings | Quantitative findings | Convergence |
|---|---|---|---|
| **Extended time spent in one location** | **Theme 1: Entrapment and escape**<br>• Feel bored (sub-theme 1.1)<br>• *Feel trapped in time and space (sub-theme 1.1).*<br>• Sense of entrapment motivated some towards recovery, others felt despondent (sub-theme 1.1).<br>• *Connection to outside world (other people, views), variety and change, and purposeful activities help to alleviate entrapment and boredom (sub-theme 1.2).*<br>• Spending too long in one room or too much time alone contributes to low mood, feel stuck (sub-theme 1.1).<br>• *Leaving the ward can relieve boredom and provide motivation, even if do not leave the hospital (sub-theme 1.3).* | **Boredom questionnaire (MSBS)**<br>• Total MSBS scores appear to be similar to population norm<br>• *But 'Time perception' subscale of MSBS differed from population norm.*<br>**Motivation questionnaire (VAS)**<br>• Participants found the environment slightly more motivating at Case 1 than Case 2, but wide variation at both cases. | **Divergent**<br>• Describe feeling bored, but MSBS scores similar to non-hospitalised population. Boredom maybe due to changes in 'Time perception'?<br>**Congruent**<br>• *Feel trapped in time & space.*<br>• Some patients feel motivated by the hospital environment, others do not.<br>**Qualitative only**<br>• *Connection to outside world, sense of variety and change, and purpose are important for emotional well-being.*<br>• Emotional well-being negatively impacted by extended time alone in bedroom.<br>• *Leaving ward can be motivating.* |
| **Presence and absence of other people–bedrooms** | **Theme 3: Rehab facility is a shared space**<br>• Difficult to find privacy on the ward–even in single-bed rooms (sub-theme 3.1).<br>• *Want privacy in bedroom, but do not want to feel isolated (sub-theme 3.1).*<br>• Roommates provide emotional support if cognitively able. If roommate is cognitively impaired then participant may feel frustrated or responsible for them (sub-theme 3.1). | **Mood questionnaire (DASS)**<br>• Other factors besides number of beds play a role in participants' mood (S4 File).<br>**Motivation questionnaire (VAS)**<br>• Motivation varied widely in single and shared rooms at both cases (S4 File).<br>**Boredom questionnaire (MSBS)**<br>• *Participants in single-bed rooms experience more boredom than in shared bedrooms (S4 File).* | **Congruent**<br>• Mood and motivation not necessarily better in single or shared bedrooms; both can lack privacy.<br>**Qualitative only**<br>• *Distinction between privacy and isolation.*<br>• Impact of roommate on emotional well-being depends on their cognitive abilities.<br>**Quantitative only**<br>• *Appear to be more bored in single-bed rooms.* |
| **Presence and absence of other people–hallways** | **Theme 1: Entrapment and escape**<br>• Seeing other people in hallway helps participants to feel less alone (sub-theme 1.2).<br>**Theme 3: Rehab facility is a shared space**<br>• Seeing patients in hallways and therapy can be upsetting but also motivating (sub-theme 3.2).<br>• *The lounge nooks at Case 2 allow for privacy, without feeling isolated (sub-theme 3.2).* | – | **Qualitative only**<br>• Hallways provide opportunity to see other people which impacts mood.<br>• *Lounge nooks are private but not isolated spaces.* |
| **Surveillance** | **Theme 1: Entrapment and escape**<br>• Bed alarms are frustrating (sub-theme 1.1).<br>**Theme 2: Power, dependency, and identity**<br>• Surveillance prompts feelings of lack of privacy and lack of autonomy; can be perceived as punishment or lack of trust (sub-theme 2.1). | – | **Qualitative only**<br>• Surveillance may cause frustration and lack of privacy, prompt feelings of powerlessness. |
| **Interest or fascination** | **Theme 1: Entrapment and escape**<br>• Some feel there's nothing interesting in the environment (sub-theme 1.1).<br>• Changes outside window provide variety, distraction, sense of progress; help motivation and boredom (sub-theme 1.2).<br>• Outside helps with boredom (sub-theme 1.3).<br>**Theme 2: Power, dependency, and identity**<br>• *Visual and aural aesthetics can help or hinder well-being. Some find clinical aesthetic dull and boring (want colour), others find it motivating or reassuring (sub-theme 2.2).* | – | **Qualitative only**<br>• Variety and change in environment (e.g., going outside or seeing outside) may help with mood, boredom, and motivation.<br>• *Ward aesthetics may help emotional well-being for some, hinder for others.* |

*(Continued)*

**Table 10.** (Continued)

| Topics | Qualitative findings | Quantitative findings | Convergence |
|---|---|---|---|
| Equipment, facilities, and technology | **Theme 2: Power, dependency, and identity in an institutional environment**<br>• Harder to personalise shared bedrooms and all bedrooms at Case 2; implications for control (sub-theme 2.3).<br>• *Sense of ownership in gym can be motivating (sub-theme 2.3).*<br>**Theme 4: The environment should be legible and patient-centred**<br>• Purposeful, modern therapy equipment provides motivation (sub-theme 4.2).<br>• Scarce, unmaintained, or inaccessible equipment, facilities, or technology can be frustrating reminders of dependency (sub-theme 4.2). | – | **Qualitative only**<br>• Sense of ownership in single and shared bedrooms is important for patients' sense of control.<br>• *Sense of ownership over equipment, facilities, and technology in gym can be motivating.*<br>• Equipment, facilities, and technology can be motivating, but they can become a reminder of dependency if they are unmaintained, lacking, or unavailable. |
| Layout and orientation | **Theme 1: Entrapment and escape**<br>• Seeing others through a window helps participants to feel connected with the outside world and less alone (sub-theme 1.2).<br>**Theme 3: Rehab facility is a shared space**<br>• Bedroom location away from communal hubs linked to isolation (sub-theme 3.1).<br>**Theme 4: The environment should be legible and patient-centred**<br>• *Participants unaware of location and existence of key spaces so feel lack of control and lack of autonomy; information about the environment can help with orientation (sub-theme 4.1).*<br>• Hard to feel oriented without landmarks on the modern, white walls at Case 2 (sub-theme 4.1). | – | **Qualitative only**<br>• Layout that facilitates connection with others (e.g., proximity to communal hubs or window with a view of people) may be beneficial for emotional well-being.<br>• *Layout and information play a role in how orientated patients feel to the space and subsequently their sense of control.*<br>• Aesthetics play a role in how oriented patients feel in a space. |
| Dimensions of the space | **Theme 4: The environment should be legible and patient-centred**<br>• Wide hallways provide motivation (sub-theme 4.2).<br>**Theme 1: Entrapment and escape**<br>• Large rooms and large windows help some patients to feel less trapped. (sub-theme 1.1).<br>• *But, for others, large rooms/windows don't help them to feel less trapped; and feelings of entrapment persisted at Case 2 despite the larger rooms and larger windows. (sub-theme 1.1).* | – | **Qualitative only**<br>• Larger rooms, windows, and hallways may help with emotional well-being.<br>• *Large rooms and windows do not eliminate sense of entrapment.* |

Font style (italicised or regular) shows which results link together across the qualitative, quantitative, and converged findings.

Findings were consistent across both cases, unless otherwise specified.

The following mark indicates no data: –

Surveillance of patients by staff and/or technology; 4) Patients' sense of interest or fascination with the space and aesthetics; 5) Equipment, facilities, and technology; 6) Layout and orientation; and 7) Dimensions of the space.

The contents of the joint display tables are briefly summarised below in a narrative integration which emphasises three environmental principles: 1) variety and interest in the environment, 2) allowing for privacy without isolation, and 3) patient-centred design. Together, these principles describe how the physical environment of inpatient rehabilitation facilities supports, or could support, patient behaviour (activity and rest), emotional well-being, and safety post-stroke.

**Variety and interest in the environment.** Converged findings suggested that patients spend extended periods of time in one, unchanging environment–their bedroom–and that

**Table 11. Comparison of the qualitative and quantitative findings that relate to patients' safety.**

| Topic | Qualitative findings | Quantitative findings | Convergence |
|---|---|---|---|
| **Extended time spent in one location** | **Theme 1: Entrapment and escape**<br>• Entrapment prompts some to seek out stimuli and so may be a safety risk (sub-theme 1.1). | **Retrospective falls record**<br>• *Most falls occurred in the bedroom.* | **Qualitative only**<br>• Extended time in bedroom may prompt risky behaviour.<br>**Quantitative only**<br>• *Most falls occur in bedroom.* |
| **Presence and absence of other people** | **Theme 2: Power, dependency, and identity in an institutional environment**<br>• Patients feel safer if they know staff will be there to help when needed (sub-theme 2.2).<br>**Theme 3: Rehab facility is a shared space**<br>• Want privacy in bedroom, but not isolation; can feel unsafe if isolated (sub-theme 3.1).<br>• *Roommates help each other (sub-theme 3.1).* | **Retrospective falls record**<br>• Most falls occur when staff not present. | **Congruent**<br>• Isolation detrimental to safety.<br>• Safer if patients can readily contact staff.<br>**Qualitative only**<br>• *Roommates may be beneficial for safety.* |
| **Surveillance** | **Theme 2: Power, dependency, and identity in an institutional environment**<br>• Surveillance may prompt some participants to sneak out or self-discharge (sub-theme 2.1). | **Retrospective falls record**<br>• Details from Case 1 indicate that proximate alarms and sensor mats are unreliable. | **Congruent**<br>• Surveillance may be detrimental to safety, or at least, not beneficial. |
| **Equipment, facilities, and technology** | **Theme 4: The environment should be legible and patient-centred**<br>• Nurse call-bell can help patients to feel safe<br>• *But nurse call-bell is unreliable if out of reach; things in the bedroom are inconveniently placed (sub-theme 4.2).* | **Retrospective falls record**<br>• Staff respond to falls risk by adjusting equipment to make patient-centred (e.g., angle of chair).<br>• *Reaching for objects can lead to falls.* | **Congruent**<br>• Patient-centred equipment and facilities may aid safety.<br>• *Inconvenient environment may hinder safety.* |
| **Dimensions of the space** | – | **Retrospective falls record**<br>• More falls in the bathrooms at Case 2 compared to Case 1, and bathrooms at Case 2 were larger. | **Quantitative only**<br>• Larger rooms may be detrimental to safety. |

Font style (italicised or regular) shows which results link together across the qualitative, quantitative, and converged findings.

Findings were consistent across both cases, unless otherwise specified.

The following mark indicates no data: –

this may have negative implications for their activity, emotional well-being, and safety. Feelings of entrapment appear to persist regardless of room size and a key means of providing a distraction from boredom or low mood is to spend time in more than one location, namely, get patients out of their bedroom. This could also benefit patients' safety since most falls occur when patients are alone in their bedroom. Our findings reveal a disconnect between patients' desires to leave their bedroom, and their actual behaviour. Our findings suggest that increasing the richness and interest of areas outside the bedroom may encourage patients to explore these spaces, thereby helping them to experience a more varied and interesting environment, encouraging activity, and supporting emotional well-being. Variation and changes in the environment over the course of a day may have the additional benefit of helping to mark time in what is otherwise a static existence.

**Allowing for privacy without isolation.** The merged results also suggest that being around other people (or at least having visual connection) is important, even when it does not result in social activity, but that patients also appreciate a sense of privacy (see 'Presence and absence of other people–bedrooms,–communal spaces, and–hallways' in all joint display tables). There appears to be an important difference for patients between choosing to be alone (privacy) and being alone without choice (isolation). To better describe this dual need for connection and privacy, a distinction is made here between a) feeling alone or isolated–defined as an awareness that there are no other people in close physical proximity, b) feeling lonely– defined as feeling unable to engage or connect with other people), and c) feeling a sense of

privacy–defined as feeling that one has control over whether other people have access to one's personal space, thoughts, feelings, and information about one's person. The issue of surveillance, for example, illustrates the distinction between wanting to be private and wanting to be alone. Although patients were wary of surveillance, many felt safer if they were not alone, and indeed, many falls could have been avoided if patients had been able to easily ask a staff member to help them to the bathroom or to pick up an object.

Connection with other people provided variety and interest for patients and it also helped to combat loneliness. Many other factors, besides the number of beds in the room, played a role in patients' experience of loneliness. These included internal factors, such as the impairments experienced by the patient and/or their roommate, and external factors, such as the position of the beds in the room, noise outside the bedroom, location of bedroom relative to communal hubs, and size and location of communal spaces. Having another person in the room is not sufficient to prevent loneliness, nor is it sufficient to promote social interaction. Similarly, being alone is not synonymous with feeling a sense of privacy. Some patients in single-bed rooms experienced a lack of privacy, especially those who had to share a bathroom, and some patients sought out places where they could be private, but not alone (e.g., cafes, hallways, waiting areas, or lounge nooks). The public nature of these spaces (ambient noise, layout, anonymity for the patient) brought a measure of privacy for the patient, without leading to isolation.

**Patient-centred design.** Our merged results show that simply providing for patient needs and having options available does not guarantee that patients will be able to exercise an informed choice or exert control. Patients need communal spaces, for example, but simply having these spaces available is not sufficient to fulfilling this need. There were several reasons why patients did not use the available communal spaces, including: not knowing they existed or where to find them, feeling that they were too crowded or difficult to access, or feeling they did not have permission to use them. These and other similar results suggest that at least three criteria need to be met in order to properly meet patients' needs and give them choice and control over their environment:

1. Provision: Appropriate options that respond to patient needs must be provided (e.g., communal spaces have to exist, must be of appropriate size, and must include all necessary equipment such as chairs, books, etc.).

2. Coherence: Patients need to know that these options are there and understand how to access them (e.g., communal spaces must be in an obvious location, be easy to find, and patients need to be given this information)–the term 'coherence' was proposed by Kaplan to describe how easily information in an environment is processed [52].

3. Convenience: Patients need to be able to access these options easily and conveniently–this will help to remind the patient that they have the right to choice and control in the environment (e.g., communal spaces must be easy for patients to get to, and easy to enter).

These three criteria can be applied to any question of patient-centred design, besides that of communal spaces. For example, the safety audit in our study revealed a need for more surfaces such as shelves or tables in the bedroom so that patients could place items within reach. In order to meet this need, extra surfaces would need to be provided (Provision), they would need to be designed and maintained in such a way that made it obvious what they were for (Coherence), and they would need to be placed in a position that was convenient for the patient, or be convenient for them to adjust or move (Convenience).

### A conceptual model of the role of the environment in patient behaviour, emotional well-being, and safety

The findings from the mixed methods analysis imply that patient behaviour (activity and rest), emotional well-being, and safety will be supported if the physical environment of rehabilitation facilities is designed according to the three environmental principles outlined above in the narrative integration. A conceptual model of this relationship is described in Fig 5. The model suggests that each of the three principles plays either a direct or indirect role in patient activity, rest, emotional well-being, and safety.

The conceptual model in Fig 5 could be applied to all aspects of the physical environment of rehabilitation facilities. To give an example, Fig 6 shows how the model could be used to inform the architectural and interior design features of the communal lounge room. If all three

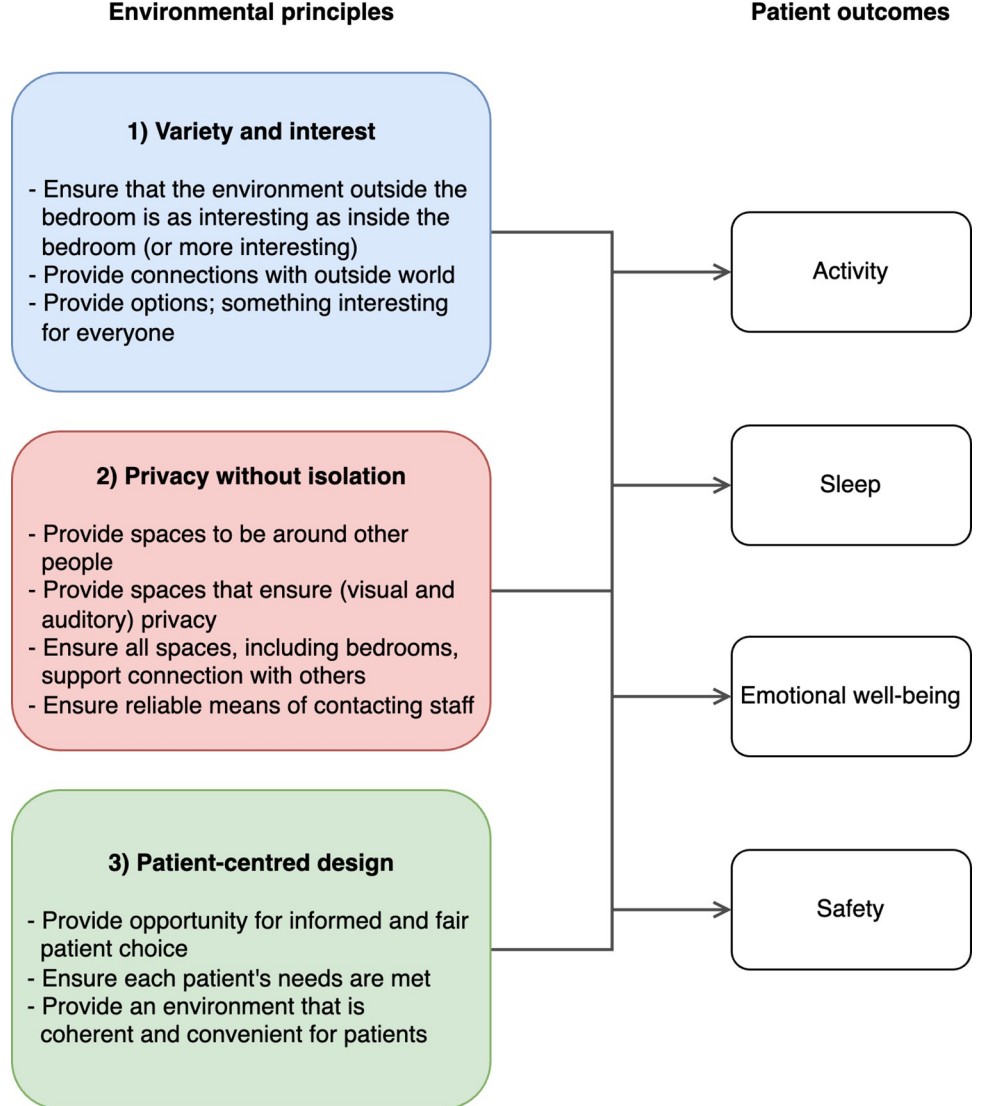

**Fig 5. A conceptual model of the environmental principles that play a role in patient behaviour (activity and sleep), emotional well-being, and safety in stroke rehabilitation.** Definitions of the terms used in this figure are provided in text and should be considered alongside this figure when applying this model in practice.

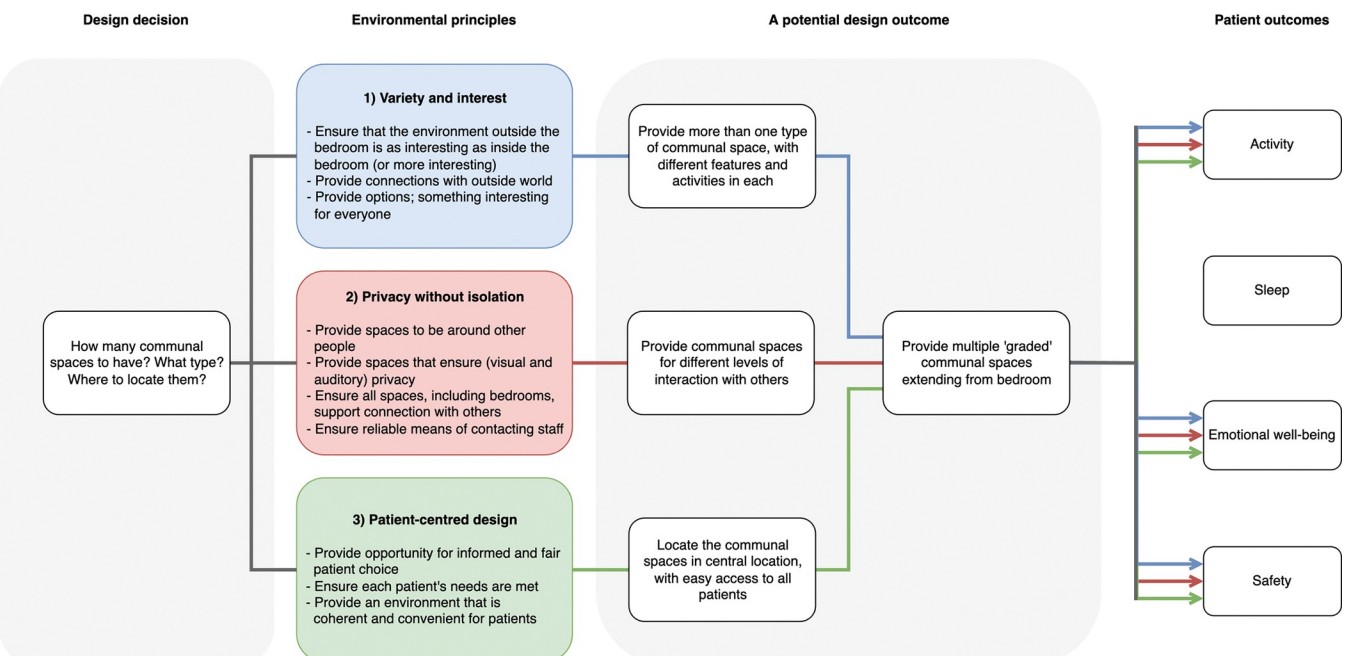

**Fig 6. An example of how the conceptual model could be used to inform the design of communal spaces in stroke rehabilitation.** The design outcome suggested here ('graded' communal spaces) is only one potential design outcome to this particular design decision; a designer might generate several stroke rehabilitation-appropriate design options by applying the ENVIRONS principles to this particular design decision.

environmental principles are applied, then a possible design outcome could be 'graded' communal spaces that begin just outside the patients' bedrooms. There could, for example, be: 1) a small gathering space immediately outside the bedroom that is shared between one or two bedrooms and that includes some resources that promote cognitive activity (e.g., books), this would connect to 2) the larger communal hallway, with some seating provided, and off this hallway could be 3) a more conventional communal lounge for larger gatherings and with more extensive resources for cognitive activity (e.g., computers, tablets, etc.).

## Discussion

The conceptual model developed in the ENVIRONS study emphasises the stroke survivor perspective and describes three over-arching, environmental principles: 1) promote variety and interest in the environment, 2) allow for privacy without isolation, and 3) patient-centred design. Through describing these principles, concepts such as privacy and patient-centred design, which are well researched in other types of healthcare environments, were defined in the context of stroke rehabilitation. The ENVIRONS conceptual model is not a prescriptive design tool. Rather, the model is designed to be used alongside the detailed design briefs that are generally provided to health facility architects and planners. Each design decision can be considered through the lens of the three principles to optimize stroke survivor activity, sleep, emotional well-being, and safety, thereby helping to inform the design of purpose-built stroke rehabilitation facilities.

We found that stroke survivors value variety, change, and purpose in their physical environment; these qualities help to provide some connection with their surroundings, relieve boredom, and lift their mood. The term 'fascination' is used in environmental psychology to describe the calm engagement and desire to explore that people feel in response to

information-rich natural or architectural environments [52, 53]. In healthcare design, the term 'positive distraction' has been used to describe aesthetic qualities in the environment, such as artwork or nature views, that patients find pleasant and interesting [54]. Critically, in a stroke rehabilitation context, these elements of 'fascination' or 'positive distraction' must be present in other parts of the ward besides the patient bedroom in order to encourage stroke survivors to spend time in communal areas and engage in the incidental physical, cognitive, and social activities that are beneficial for recovery.

Findings from the qualitative interviews indicate that what constitutes an interesting environment (or a fascinating environment, or a positive distraction) may vary between patients–aesthetics that are uplifting and engaging for some stroke survivors may feel depressing and institutional for others (see 'Extended time in one location' and 'Interest and fascination', Table 10). Versatility in the rehabilitation environment is therefore key in order to accommodate all tastes and create interest and incentives for all stroke survivors to explore beyond their bedroom, as highlighted in previous research of what is important in rehabilitation environments [23]. Interior design could, for example, provide options for a space to be either bright and colourful or neutral and soft depending on the preferences of the patient. Kevdzija and Marquardt, who conducted an exploratory study of stroke rehabilitation environments in Germany contemporaneously to the ENVIRONS study, similarly concluded that a wide variety of common spaces are required to meet the varied needs and interests of all stroke survivors [55].

The dual need for both connection and privacy in stroke facilities has been raised in previous research [56]. Akin to our finding that stroke survivors want to be private but not isolated, and the importance of the lounge nooks in the hallways at Case 2, stroke survivors in the Kevdzija and Marquardt study reported that they seek out quiet, common spaces where they can be in public but undisturbed and the importance of hallways as a common space [17, 55]. The issue of privacy is not unique to patients in shared bedrooms–we found that stroke survivors in both single and shared bedrooms struggled to find privacy on the ward, and those sharing a bathroom found this especially hard. It is well established that patients in rehabilitation, and especially in single-bed rooms, may feel lonely, and many participants in shared bedrooms appreciated the company of roommates [57–60]. Participants in shared bedrooms at both cases described taking care of their roommates and helping to keep each other emotionally and physically safe (Theme 3). As in the present study, previous studies have similarly found that some rehabilitation patients will forgo a single-bed room in order to avoid isolation [61].

Privacy has long been acknowledged as an important issue in health care ethics [62], but there is no unified definition of what constitutes privacy in this context [63]. The definition adopted in this study–feeling that one has control over whether other people have access to one's personal space, thoughts, feelings, and information about one's person–draws from Altman's definition of privacy, which emphasises the importance of one's volition and control over whether other people have access to one's self [64], and Burgoon's definition, which emphasises that privacy is about more than just proximity to others, it is also about choosing whether to interact, and about the sharing of personal thoughts and information [65]. Privacy in the healthcare environment is clearly more than just the issue of single versus shared bedrooms, it is also about where and how staff conduct consultations or discuss confidential patient information, and it is about whether patients feel that they have a 'territory' or personal space (akin to our concept of ownership, see 'Equipment, facilities, and technology', Table 10) [63, 66, 67].

The extent to which stroke survivors feel their privacy is being respected may influence how much they trust staff, and subsequently how much they communicate and disclose [67]. This has implications for our findings regarding surveillance, which suggest that stroke survivors feel that surveillance is an invasion of their privacy, that it makes them feel powerless and

frustrated, and that it jeopardises the trust between patient and staff (see 'Surveillance', Table 10). In addition, findings from our falls audit reflect previous research which found no support for physical restraints (e.g., motion sensors or bed alarms) to reduce patient falls [68]. Indeed, efforts to reduce the use of these restraints have resulted in a reduction of patient falls and injuries [69, 70]. Surveillance, including bed alarms and being watched, contributed to participants' feeling that they did not have permission to go anywhere except their bedroom, and so may have contributed to low levels of activity. Interventions such as hourly nursing rounds have been trialled in a number of settings and have led to moderate improvements in patients' perception of nurse responsiveness and a moderate reduction in patient falls [71]. Physical environments that minimise patient isolation could reduce the need for such systems. Just as the distinction between privacy and isolation lies in patient choice, there is a similarly important distinction between choosing to be watched (asking for observation or using the call-bell) and being watched without choice (surveillance).

For an environment to be patient-centred it must meet the needs of the patient, or be adaptable to meet these needs, empower the patient, and give them control [72, 73]. Our findings help to define a patient-centred rehabilitation environment as one that provides everything the stroke survivor needs in a manner that is coherent and convenient for them, thereby enabling them to exercise informed choice and control over their environment. Other authors have similarly concluded that stroke survivors are more likely to use communal lounges that are located a short distance from their bedroom because they are easier to find (coherent) and access independently (convenient) than lounges located further away [17, 18, 55]. Our findings suggest that environments that provide only the means to meet patient needs, without making these means coherent or convenient for the patient, risk becoming detrimental to patient behaviour, emotional well-being, and safety, rather than beneficial. Aspects of the environment that were not coherent or convenient for stroke survivors (e.g., poorly placed timetables, locked doors to balconies, communal lounges that were difficult to find, and even power socket placement, see Theme 4, Table 5) made participants feel that they had little choice or control in their environment, and reminded them of their lack of ownership over the environment, discouraging independent activity. The experiences described by participants suggest that patient-centred design should be applied to all aspects of the environment, as even the smallest feature or design choice can impact a patient's sense of ownership.

Congruency between the qualitative and quantitative data in the joint display tables helped to confirm findings, and the divergent and unique data helped to explain findings, expanding on the congruent results. The convergent mixed methods approach provided greater insights and a deeper understanding of this complex topic than either the qualitative or quantitative methods alone.

The broad eligibility criteria and flexible participation options allowed for the intended, diverse sample of participants, including people with language, cognitive, and/or mobility impairments. Good approach and consent rates were achieved, and many patient experiences were common to participants at both cases, to participants in single and shared bedrooms, and to participants with different types of impairments, which speaks to the relevance of our findings. People with *severe* language or cognitive impairments (e.g., global aphasia) were, however, not included, and so their experiences may not be represented in the study findings. In addition, our data were collected in only two rehabilitation facilities and stroke survivor experiences may be different in other building types.

As with all overt observation, the behavioural mapping conducted in this study may have influenced participant behaviour. Still, on balance, overt observation was preferred, as covert observation (observing without consent) is ethically questionable, especially in a healthcare context [74]. Our behavioural mapping was conducted in snapshot observations (five-seconds

at each epoch), rather than continuously throughout the observation period. This had multiple benefits, including being less obtrusive for participants and allowing observers to observe multiple participants sequentially during each epoch. A limitation the snapshot method, however, is that the observed activity may not have been representative of the participant's activity during the whole epoch.

The extent of missing questionnaire data highlights the challenges of data completeness in this population [75]. Most missing data were due to participants with language impairments not feeling confident to complete the questionnaires, even with support. The adapted Questionnaire Booklet was useful but was evidentially not sufficient as the response scales and the wording of the questions could not be changed without jeopardising the validity of the measures. All participants were, however, able to complete the interview, which is testament to the feasibility of using a flexible interview design with this cohort.

## Conclusion

The unique needs and clinical priorities of stroke rehabilitation should be reflected in the physical environment of inpatient rehabilitation facilities to facilitate optimal experience and care. The ENVIRONS model describes three design principles which, together, suggest how these unique needs and priorities can be met. To design stroke rehabilitation spaces that are fit for purpose, architects and planners can filter their design choices through the lens of these principles, considering the extent to which each aspect of their design promotes variety and interest in the environment, allows for privacy without isolation, and provides a patient-centred environment that is coherent and convenient for stroke survivors.

## Supporting information

**S1 Table. Description of the environments of the two cases in the ENVIRONS study.**
(DOCX)

**S2 Table. Illustrative quotes from the walk-through semi-structured interviews.** Minimum data set for the qualitative data from the ENVIRONS study.
(DOCX)

**S1 Fig. Topics describing the important aspects of the qualitative and quantitative findings in the ENVIRONS study.**
(DOCX)

**S1 File. Healthcare design checklists used in the ENVIRONS study: Methods & Results.**
(DOCX)

**S2 File. Semi-structured walk-through interview used with stroke survivors in the ENVIRONS study.**
(DOCX)

**S3 File. Research results summary sent to ENVIRONS study participants.**
(PNG)

**S4 File. Further quantitative findings from the ENVIRONS study.**
(DOCX)

## Acknowledgments

We thank Julie Luker, Sarah-May Blaschke, and Dennisse Bonanno who reviewed the study protocol and procedures and participated in the peer review workshop, contributing their expertise to review and refine the qualitative results. Erin Godecke and Sarah D'Souza provided essential advice regarding adaptation of study documents for participants with aphasia. We thank all staff at both case study sites, and all participants for so generously sharing their time and experiences.

## Author Contributions

**Conceptualization:** Ruby Lipson-Smith, Heidi Zeeman, Julie Bernhardt.

**Data curation:** Ruby Lipson-Smith.

**Formal analysis:** Ruby Lipson-Smith.

**Investigation:** Ruby Lipson-Smith.

**Methodology:** Ruby Lipson-Smith, Heidi Zeeman, Julie Bernhardt.

**Resources:** Leanne Muns, Faraz Jeddi, Janine Simondson.

**Supervision:** Heidi Zeeman, Julie Bernhardt.

**Visualization:** Ruby Lipson-Smith.

**Writing – original draft:** Ruby Lipson-Smith.

**Writing – review & editing:** Ruby Lipson-Smith, Heidi Zeeman, Leanne Muns, Faraz Jeddi, Janine Simondson, Julie Bernhardt.

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
