## [Decision Letter · Decision Letter 0]

19 Dec 2022

PONE-D-22-29523The role of the physical environment in stroke recovery: Evidence-based design principles from a mixed-methods multiple case studyPLOS ONE

Dear Dr. Ruby Lipson-Smith,

Thank you for submitting your manuscript to PLOS ONE. After careful consideration, we feel that it has merit but does not fully meet PLOS ONE’s publication criteria as it currently stands. Therefore, we invite you to submit a revised version of the manuscript that addresses the points raised during the review process.

We look forward to receiving your revised manuscript.

Kind regards,

Won-Seok Kim

Academic Editor

PLOS ONE

Journal Requirements:

"RLS was supported by a Research Training Program PhD scholarship from the Australian federal government. JB is supported by a NHMRC Fellowship (RF1058635). The Florey Institute of Neuroscience and Mental Health acknowledges the support from the Victorian government and in particular the funding from the Operational Infrastructure Support Grant."

Reviewers' comments:

Reviewer's Responses to Questions

**Comments to the Author**

1. Is the manuscript technically sound, and do the data support the conclusions?

Reviewer #1: Yes

2. Has the statistical analysis been performed appropriately and rigorously? 

Reviewer #1: Yes

3. Have the authors made all data underlying the findings in their manuscript fully available?

Reviewer #1: Yes

4. Is the manuscript presented in an intelligible fashion and written in standard English?

Reviewer #1: Yes

5. Review Comments to the Author

Reviewer #1: This study explores the effect of physical surroundings on patients' behaviour, emotional well-being and safety, from the patients' perspective. This topic is interesting and clinically relevant since patients' active behaviour is important for stroke recovery. The authors applied a mixed-methods multiple case study which integrates qualitative and quantitative results to attain more reliable results. I have several questions and comments which are described below.

1. (P. 7) How were observations made every 10 minutes? Did observers follow the patients all the time? If so, is there a possibility that their presence affected the behaviour of the patients?

2. (P. 10) How were divergent findings in qualitative and quantitative analysis treated, interpreted and explained? If most results were congruent, what additional benefits and information were obtained from quantitative results?

3. (P. 21, 33) The authors concluded that extended time in bedroom is detrimental to safety (Table 11). The reasons for most of the falls happening in the bedroom may have been simply because most of the time patients were in the bedroom. The causality is not clear.

4. (P. 44) The conclusion seems to emphasize the architectural design of stroke rehabilitation facilities. Some aspects of the patients' needs may be addressed by education and behavioural interventions rather than changes in the actual physical surroundings. (For example, providing information and education to ease access to existing communal spaces.)

6. PLOS authors have the option to publish the peer review history of their article (what does this mean?). If published, this will include your full peer review and any attached files.

Reviewer #1: No

---

## [Author Response · Author response to Decision Letter 0]

3 Jan 2023

Comments from academic editor

The manuscript has been reviewed for style requirements. Files have been re-name according to the requirements (space removed) and re-uploaded. We referred to the 2021 versions of the style requirements accessed via the PLoS ONE website (noting that the links above go to 2017 versions).

Tables 8 to 11 of the submitted manuscript include coloured text, as we consider this essential for interpretation of the table contents; the colour links the relevant qualitative findings and quantitative results and shows how they have been converged together in the mixed methods analysis. We note that PLoS ONE normally requires that no coloured text is used in Tables (black only), and we ask for an exception in this instance, at your discretion.

The ‘Funding Information’ has been updated to match the ‘Financial Disclosure’ statement (provided in the cover letter). Grant numbers have been added where applicable.

"RLS was supported by a Research Training Program PhD scholarship from the Australian federal government. JB is supported by a NHMRC Fellowship (RF1058635). The Florey Institute of Neuroscience and Mental Health acknowledges the support from the Victorian government and in particular the funding from the Operational Infrastructure Support Grant."

The updated financial disclosure statement has been included in the cover letter for the resubmission.

The reference list has been reviewed. None of the cited papers have been retracted.

Reviewers' comments:

Reviewer #1: This study explores the effect of physical surroundings on patients' behaviour, emotional well-being and safety, from the patients' perspective. This topic is interesting and clinically relevant since patients' active behaviour is important for stroke recovery. The authors applied a mixed-methods multiple case study which integrates qualitative and quantitative results to attain more reliable results. I have several questions and comments which are described below.

1. (P. 7) How were observations made every 10 minutes? Did observers follow the patients all the time? If so, is there a possibility that their presence affected the behaviour of the patients?

We thank the reviewer for this question as it highlighted to us that more detail was needed in the methods section and that additional strengths and limitations should be included in the discussion. We have made these amendments as detailed below.

Observers did not follow the participants all the time, rather, they observed participants for up to five seconds during each 10-minute interval (epoch). During these five second observations, the observer aimed to be discrete and unobtrusive, remaining unnoticed by the participant where possible (e.g., glancing into the participant’s bedroom while walking past the open bedroom door). Outside of these five seconds, the observer attempted to remain out of sight of the participant, but in a position that would allow the observer to notice if the participant moved to a new location (e.g., observer stationed in the ward corridor with a view of people entering and exiting the participant’s bedroom). To help the observer anticipate the whereabouts of the participants, the observer familiarised themselves with the participant’s timetable at the start of the observation day and used this as a guide as to where to expect the participant to be at any given time (bedroom, therapy room, etc.). At each epoch, the participant was presumed to have participated in that activity for the preceding ten minutes. 

Line 159 on page 7 (Methods section) has been amended to read: “Observations were made for five seconds every 10 minutes over a nine-hour period on any weekday, generating 54 discrete observation epochs per participant including three randomly allocated observer breaks. At each epoch, the observer recorded the participant’s physical, cognitive, and social activity, location, and who they were with. At each epoch, the participant was presumed to have participated in that activity for the preceding ten minutes.”

This method had strengths and limitations. Strengths included that it allowed the observer to observe up to three participants on a single observation day, rotating between participants during each epoch. In addition, this method was less obtrusive for the participant than continual observation, as they were not being watched all the time. Participants were, however, aware that they were being observed, even if they could not see the observer, as they had provided consent for observation to occur on a given day, and, as with all overt observation, this may have influenced their behaviour. Another limitation of this method is that the five second observation may not have been representative of the participant’s activity during the whole 10-minute epoch. 

A section has been added to the Discussion (page 45) to detail the strengths and limitations of this method: “As with all overt observation, the behavioural mapping conducted in this study may have influenced participant behaviour. Still, on balance, overt observation was preferred, as covert observation (observing without consent) is ethically questionable, especially in a healthcare context (Lipson-Smith and McLaughlan, 2022). Our behavioural mapping was conducted in snapshot observations (five-seconds at each epoch), rather than continuously throughout the observation period. This had multiple benefits, including being less obtrusive for participants and allowing observers to observe multiple participants sequentially during each epoch. A limitation the snapshot method, however, is that the observed activity may not have been representative of the participant’s activity during the whole epoch.”

References: Lipson-Smith R, McLaughlan R. Mapping Healthcare Spaces: A Systematic Scoping Review of Spatial and Behavioral Observation Methods. Health Environments Research & Design Journal (HERD). 2022;15(3).

2. (P. 10) How were divergent findings in qualitative and quantitative analysis treated, interpreted and explained? If most results were congruent, what additional benefits and information were obtained from quantitative results?

All the divergent findings (where qualitative and quantitative findings did not agree with each other) were included in the joint display tables in the Results section – see fourth column ‘Convergence’ in Tables 8 to 11. The convergent and divergent findings were given equal consideration, and an explanation was provided for all divergent findings when synthesising the results. These explanations were included in the joint display tables, or in the narrative integration, as appropriate. We thank the reviewer for noticing that we did not specify in the Methods section how divergent findings would be treated. We have made amendments to clarify this, see Page 11, line 235: “The results of the joint display tables – including the convergent, divergent, and unique findings – were then summarised in a narrative integration which provided an overarching explanation of the converged findings. This integration is expressed as a conceptual model describing the role of the physical environment of inpatient rehabilitation facilities in patients’ behaviour, emotional well-being, and safety after stroke, thereby addressing the aim of this study.”

While most of the results were congruent (see Tables 8 to 11), some of the divergent findings were especially revealing and were critical to the narrative integration and the resulting conceptual model. We provide example below of the impact of divergent findings from Table 8. 

Page 36, line 518 of narrative integration: “Our findings reveal a disconnect between patients’ desires to leave their bedroom, and their actual behaviour…” These divergent findings are listed in Table 8 – patients wanted to go to outdoor and communal spaces but did not spend time in these spaces despite these spaces being available. This divergent finding was critical to our conceptual model. Throughout the narrative integration, we provide possible explanation (drawn from our other converged findings) as to why this disconnect, or divergence, may occur, e.g., page 38, line 557 of the narrative integration: “Patients need communal spaces, for example, but simply having these spaces available is not sufficient to fulfilling this need. There were several reasons why patients did not use the available communal spaces, including: not knowing they existed or where to find them, feeling that they were too crowded or difficult to access, or feeling they did not have permission to use them.”

Where divergent findings have not been explored in detail in the narrative integration, possible explanations for the divergence have been included in the joint display table. E.g., Table 9: “Divergent: Patients feel that sleep could be improved in rooms with fewer beds, but quantitative results indicate that sleep is impacted in single-bed rooms (may be due to differences in clinical profile).”

In addition to the divergent findings, there were also unique quantitative findings, where only quantitative findings were available (see Table 8 to 11). These, too, were examples of where additional benefits and information were obtained from quantitative results. 

3. (P. 21, 33) The authors concluded that extended time in bedroom is detrimental to safety (Table 11). The reasons for most of the falls happening in the bedroom may have been simply because most of the time patients were in the bedroom. The causality is not clear.

We agree that these findings are not sufficient to imply causality. We have amended this section of Table 11 to be two unique findings (qualitative and quantitative), rather than one congruent finding. This section of Table 11 (page 34) now reads:

“Qualitative only

• Extended time in bedroom may prompt risky behaviour.

Quantitative only

• Most falls occur in bedroom.”

4. (P. 44) The conclusion seems to emphasize the architectural design of stroke rehabilitation facilities. Some aspects of the patients' needs may be addressed by education and behavioural interventions rather than changes in the actual physical surroundings. (For example, providing information and education to ease access to existing communal spaces.)

We thank the reviewer for this comment. We wholeheartedly agree that changes to the social environment (e.g., service/education/practice) can be considered alongside changes to the physical environment, and that changes to the social environment may at times be sufficient. We have taken the stance that the social and physical environment are interconnected, and that they interact with each other and adapt accordingly (see Page 2, line 51). Having said this, while we do acknowledge the interconnectedness between the social and physical environment, our research questions focus on the physical environment in particular – we did not collect data specific to the social environment or the educational and behavioural interventions that could be conducted. To clarify that our focus is on the physical environment, we have added the word ‘physical’ to our aim (page 3), which now reads: “with the aim to develop a conceptual model to inform the physical design of purpose-built facilities for inpatient stroke rehabilitation.” 

In accordance with our research question and aim, our main findings (conceptual model of design principles) are regarding the physical environment and the changes that could be made to the architectural design of stroke rehabilitation facilities, and so our conclusion reflects this too (page 44). We are, however, currently conducting a much larger project on the design of stroke rehabilitation facilities which explicitly explores both the design of the physical environment and the design of the service (social environment). Publications are currently in preparation for this larger project.

---

## [Decision Letter · Decision Letter 1]

6 Jan 2023

The role of the physical environment in stroke recovery: Evidence-based design principles from a mixed-methods multiple case study

PONE-D-22-29523R1

Dear Dr. Ruby Lipson-Smith

We’re pleased to inform you that your manuscript has been judged scientifically suitable for publication and will be formally accepted for publication once it meets all outstanding technical requirements.

Kind regards,

Won-Seok Kim

Academic Editor

PLOS ONE

Reviewers' comments:

Reviewer's Responses to Questions

**Comments to the Author**

1. If the authors have adequately addressed your comments raised in a previous round of review and you feel that this manuscript is now acceptable for publication, you may indicate that here to bypass the “Comments to the Author” section, enter your conflict of interest statement in the “Confidential to Editor” section, and submit your "Accept" recommendation.

Reviewer #1: All comments have been addressed

2. Is the manuscript technically sound, and do the data support the conclusions?

Reviewer #1: Yes

3. Has the statistical analysis been performed appropriately and rigorously? 

Reviewer #1: Yes

4. Have the authors made all data underlying the findings in their manuscript fully available?

Reviewer #1: Yes

5. Is the manuscript presented in an intelligible fashion and written in standard English?

Reviewer #1: Yes

6. Review Comments to the Author

Reviewer #1: (No Response)

7. PLOS authors have the option to publish the peer review history of their article (what does this mean?). If published, this will include your full peer review and any attached files.

Reviewer #1: No

---

## [Editor Report · Acceptance letter]

12 Jan 2023

PONE-D-22-29523R1 

The role of the physical environment in stroke recovery: Evidence-based design principles from a mixed-methods multiple case study 

Dear Dr. Lipson-Smith:

I'm pleased to inform you that your manuscript has been deemed suitable for publication in PLOS ONE. Congratulations! Your manuscript is now with our production department. 

Kind regards, 

on behalf of

Dr. Won-Seok Kim 

Academic Editor

PLOS ONE